# Disentangling Sequence Memorization and General Capability in LLMs

## Abstract

Verbatim memorization in large language models remains a persistent and unsolved challenge, raising critical concerns for privacy, copyright, and responsible deployment. Existing research suggests that effective unlearning requires targeting the specific neurons responsible for memorization, as broad model updates fail to erase content reliably. However, we show that even these approaches rest on a flawed premise. Through controlled experiments, we demonstrate that memorized sequences are not naturally isolated to specific neurons during training, except in cases where the sequences are highly atypical. In this work, we put forward a new training paradigm that attempts to **isolate memorization to specific neurons by design**. The core challenge is that gradients from the repeated sequences entangle both "generalizing" features that improve general capability, in addition to sequence-specific memorization. We show that a simple change to standard training can implicitly disentangle these by leveraging metadata that identifies repeated sequences. We verify the efficacy of our method (`SeqTD`) in a proof-of-concept natural language setting and unveil the mechanism by which this disentanglement is possible through the training dynamics of memorization. We conclude by discussing the practical considerations of the deployment of `SeqTD` and highlight potential avenues for incorporating it into large-scale settings.

## 1 Introduction

Large language models are known to memorize sequences that they observe frequently during pretraining (Carlini et al., 2023; Nasr et al., 2023). As a result, it remains possible to extract private information, copyrighted content, and infer the membership of sequences in the training dataset. Due to the legal and ethical risks of these possibilities, significant research has investigated techniques for identifying and removing such memorized sequences (Maini et al., 2024; Patil et al., 2023; Barbulescu & Triantafillou, 2024). Extensive prior research has aimed to identify the parts of a model responsible for memorization and selectively remove them (Chang et al., 2024b; Chen et al., 2024; Bayazit et al., 2024; Guo et al., 2024). These methods rest on a critical assumption: that memorization is confined to specific neurons that play little role in broader language modeling. But does standard training actually produce such neatly isolated memorization neurons? Surprisingly, this fundamental question remains largely unexplored.

In Section 3, we perform a controlled study and find that existing localization methods struggle when memorized sequences are *typical* (linguistically similar to the broader training distribution). Many undesirable cases of memorization fall in this class: copyrighted books and articles generally include broadly applicable linguistic patterns. Our findings challenge the underlying premise of post-hoc localization—in many cases, cleanly isolated memorization neurons may not exist.

If standard pretraining techniques do not induce isolation, are there alternative strategies that promote it? A potentially "obvious" approach is to route repeated sequences to their own set of neurons, essentially creating memorization neurons by design. In Section 4, we show a critical flaw of this approach: it inhibits learning general linguistic patterns across sequences, undermining the fundamental goal of pretraining. Thus, it appears some neurons must be "shared" (allowed to learn from all sequences) to maximally pick up generalizing patterns. This presents a dilemma: if we allow shared neurons, they could implement memorization—leading us back to the failure mode of standard training. Could we somehow decompose what a model learns into "generalizing" and

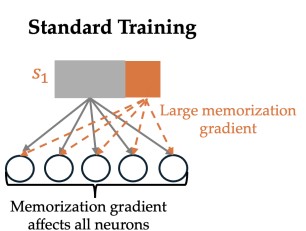
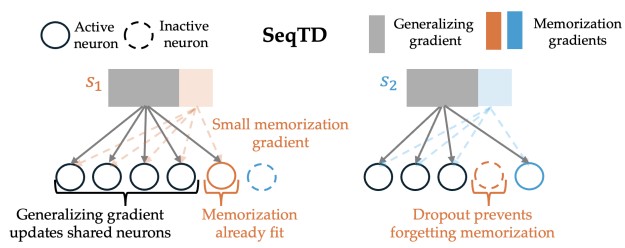

Figure 1: **Conceptual Intution of `SeqTD`**. We conceptually partition the learning signal from each example to into "generalization" and "memorization" components. On the left, we show that standard training can store memorization signal in any neurons. In `SeqTD`, we provide a set of memorization neurons which are shielded from forgetting induced by other examples. As a result, (a) memorization accumulates in these neurons and (b) once these neurons fit the memorized sequence well, memorization is no longer reinforced in shared neurons.

"memorizing" components and direct them to different neurons? This is a daunting task; it is difficult to even precisely delineate these components. However, we show that it is indeed possible, by carefully leveraging the training dynamics of memorization.

We introduce *Sequence-Tied Dropout* (`SeqTD`) which splits hidden-layer neurons in MLP layers of transformers into two groups: a pool of "shared" neurons that all examples can update, and a set of "memorization" neurons that each repeated sequence consistently activates (Section 5). By ensuring each repeated sequence drops out *all but a fixed subset* of the memorization neurons, we let memorization accumulate in that subset while shielding shared neurons from repeatedly having memorization reinforced. This design is inspired from Maini et al. (2023), and leverages the learning-and-forgetting cycles of memorization (Toneva et al., 2018): repeated text is systematically "forgotten" in the shared parameters due to interference from other examples, while memorization neurons that only see a small subset of data become stable long-term storage. Crucially, this allows partial parameter sharing so that repeated text can contribute general linguistic signals to the model.

On a modified TinyStories pretraining setup, we show that `SeqTD` isolates memorization significantly better than post-hoc localization. After training, simply zeroing out the memorization neurons suffices to "unlearn" repeated sequences without noticeably harming the model's performance on other data (Section 5.1). We then investigate the two main practical requirements for applying `SeqTD`: the accuracy of sequence metadata and model size (Section 5.2). We find `SeqTD` is capable of withstanding some amount of noise in sequence metadata (up to 10%) and can isolate memorization across a wide range of model sizes. Finally, we investigate the mechanism by which `SeqTD` isolates memorization and provide experimental evidence of the role of learning-forgetting dynamics in its success (Appendix F). Ultimately, we present a principled approach for the intricate, yet crucial puzzle of disentangling memorization from the general capabilities of LLMs.

## 2 RELATED WORKS

Research on unlearning memorized information in neural models includes exact methods like SISA (Bourtoule et al., 2021) and approximate post-hoc approaches (Triantafillou et al., 2023). With increasing concerns about memorization in large language models (Carlini et al., 2023; Nasr et al., 2023), recent methods either adjust model parameters (Thudi et al., 2022; Liu et al., 2022; Zhang et al., 2024; Yao et al., 2024) or identify and remove responsible components (Chang et al., 2024b; Chen et al., 2024; Stoehr et al., 2024; Bayazit et al., 2024; Guo et al., 2024). However, these methods often degrade overall model performance (Maini et al., 2023; Zhang et al., 2024). We propose a pretraining technique to remove memorized content while preserving model capabilities.

## 3 PITFALLS OF POST-HOC LOCALIZATION

Prior works attempt to measure the contribution of each neuron to memorization and subsequently remove the top scoring ones. This assumes sequence memorization is implemented by some subset

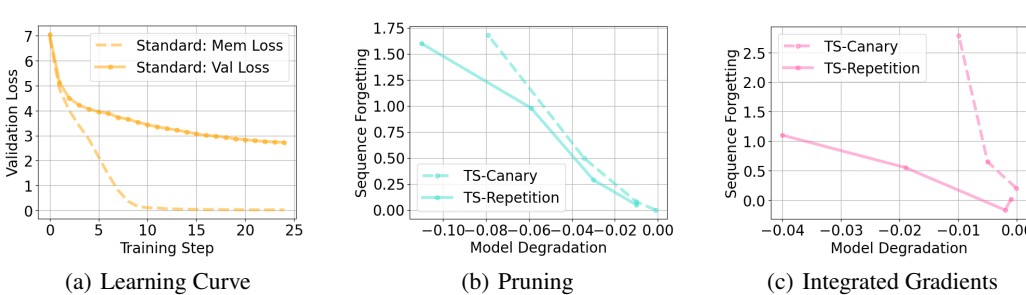

(a) Learning Curve        (b) Pruning        (c) Integrated Gradients

Figure 2: **Study of Localization** (a) Loss curve when training on `TS-Repetition`. Memorization decreases alongside the validation loss. (b) We plot the unlearning-model degradation tradeoff of pruning by varying the number of dropped out neurons and demonstrate the method struggles to unlearn sequences of both kinds (c) Integrated gradients mitigates model degradation in both cases but struggles with removing typical sequences.

of neurons and that these neurons must also don't contribute to the model's general capabilities. In this section, we study whether standard training naturally satisfies these requirements by examining the performance of localization methods in a controlled setup. We defer additional details of the two methods we study (pruning and integrated gradients) to Appendix C.

## 3.1 EXPERIMENTAL SETTING

We train models on two controlled settings designed to induce different types of memorization: highly atypical canaries and typical sequences that resemble normal text.

**Datasets..** We conduct experiments using a subset of TinyStories (Eldan & Li, 2023) to simulate real-world memorization. In the `TS-Repetition` setting, 100 stories are repeated 128 times, making memorized sequences typical. In `TS-Canary`, random token sequences (Canaries) are appended to 100 stories and repeated 128 times, creating more atypical sequences. Both setups include 20,000 un-repeated TinyStories sequences.

**Evaluation Metrics.** We measure **sequence forgetting** as the difference in loss on repeated sequences before and after localization and dropout (higher is better). We measure the **model degradation** as the difference between the validation loss before and after removal (higher is better).

## 3.2 EMPIRICAL OBSERVATIONS

We show the results of our analysis in Figure 2. Both post-hoc methods achieve limited success and struggle particularly to remove typical memorized sequences from `TS-Repetition`.

**Memorization and Generalization Occur Simultaneously.** In Figure 2(a), we plot the validation and memorization of a model trained on `TS-Repetition`. We see that the loss on repeated sequences and the validation set descend simultaneously. Our observations are supported by prior works, such as Tirumala et al. (2022), that observe memorization of sequences occurs prior to overfitting. The simultaneous learning of memorization and generalization illustrates the challenge of avoiding memorization: simply removing repeated sequences can harm model capability.

**Localization Methods Achieve Partial Success.** In Figure 2(b) we show the trade-off in sequence forgetting and model degradation of pruning. In both settings dropping out the identified neurons leads to an increase in the memorized sequence's loss, suggesting some success in localization. There are similar trends in Figure 2(c) for integrated gradients, although we observe it generally produces less model degradation than pruning. We see that integrated gradients is less effective in removing memorization in `TS-Repetition`, while being effective in `TS-Canary`.

**Typical Sequence are Difficult to Remove Post-hoc.** Across both methods, we find that applying post hoc methods to `TS-Repetition` results in greater model degradation than `TS-Canary`. This difference is particularly pronounced for integrated gradients. Recall that the memorized se-

quences in `TS-Repetition` are "typical"– similar to the non-repeated training data and the validation set. Our results suggest the memorization of typical sequences may not be isolated in neurons.

In summary, our controlled study suggests that while highly *atypical* memorized sequences appear to be isolated by standard training, the same is not true for more typical sequences. Our findings challenge the feasibility of simply removing memorization post-hoc and suggest the need to explicitly promotes isolation during pre-training.

## 4 INSUFFICIENCY OF ENFORCING LOCALIZATION

Previously, we saw removing memorization neurons post-training is challenging. A more direct approach is to enforce their creation during pretraining. This can be done by restricting repeated sequences to update a separate, known set of neurons (Gradient Masking)—seemingly ensuring the existence of known memorization neurons by design. We describe full details of our implementation in Appendix D.

However, we find that this rigid approach **both** (a) hinders learning general features across sequences and (b) fails to truly isolate memorization. This reveals that simply "forcing" localization can do more harm than good: memorization can continue to become entangled with general capability, while desirable cross-sequence learning is inhibited.

**Gradient Masking Hinders Cross-Sequence Learning.** We observe that the performance of gradient masking is inferior to a standard model, even before memorization neurons are removed (Figure 3). This observation renders gradient masking impractical, as it significantly worsens the model's general capabilities. This finding suggests that it is essential for some "shared" neurons to be updated by all sequences to aggregate general features.

**Gradient Masking Does Not Fully Isolate Memorization.** In Figure 3, we see removing memorization neurons further degrades validation performance. This indicates that gradient masking also fails to fully isolate memorization from general capabilities. Even though the dropped out neurons only received "memorization" gradients, the forward pass leaks activations between memorization neurons and the rest of the model. As a result, general capabilities become sensitive to the removal of memorization neurons during training.

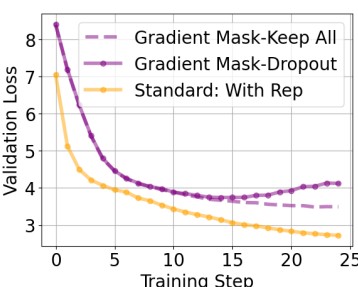

Figure 3: **Impact of Gradient-Masked Training**. We compare the validation loss of gradient-masked training with (Gradient Mask-Dropout) and without (Gradient Mask-Keepall) memorization neurons removed to a standard training run (Standard). Gradient-masked training achieves a significantly worse validation loss and dropping out memorization neurons further degrades validation performance as training progresses.

In summary, shared neurons are necessary to facilitate the learning of general linguistic capabilities across all sequences. Moreover, isolation must go beyond simply forcing memorization to separate neurons. Can we simultaneously resolve both challenges? Next, we show how carefully leveraging the dynamics of memorization can cause isolation to *naturally arise*, even with shared neurons.

## 5 SEQUENCE-TIED DROPOUT (SEQTD)

To address the challenge of isolating memorization and generalization signal in LLMs, we propose a novel pretraining strategy for transformers called Sequence-Tied Dropout (SeqTD), to simultaneously achieve two goals: (1) **Preserve cross-sequence learning** and (2) **Enforce effective isolation**.

In standard training, memorized sequences undergo learning and forgetting cycles, reinforcing memorization throughout the model. To counter this, we propose *Sequence-Tied Dropout* (SeqTD), which assigns each sequence a fixed subset of memorization neurons. By limiting their updates to fewer sequences, these neurons retain memorization while preventing reinforcement elsewhere

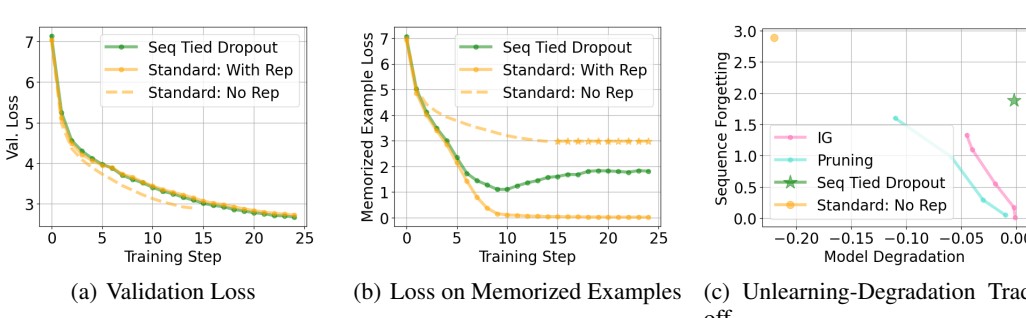

(a) Validation Loss      (b) Loss on Memorized Examples      (c) Unlearning-Degradation Trade-off

Figure 4: **Performance of `SeqTD`** (a) We find that `SeqTD` achieves a comparable validation loss to a normally trained model on `TS-Repetition`, outperforming a model trained without repeated sequences. (b) We show the loss of `SeqTD` on the repeated sequences, showing that it memorizes significantly less than a normally trained model. (c) We compare the sequence forgetting-model degradation tradeoff of `SeqTD`, relative to the post-hoc methods tested in Section 3, finding `SeqTD` outperforms both. We compute the model degradation for `SeqTD` and Standard: No Rep as the difference in the validation loss relative to a standard trained model on `TS-Repetition`.

in the model. This ensures memorization accumulates in designated neurons while generalization remains unaffected. We provide full implementation details in Appendix E.

`SeqTD` extends prior work on localizing memorization (Maini et al., 2023) in three key ways. Firstly, we position the localization of memorization in the realistic scenario of *typical* sequence memorization (like copyrighted books), as opposed to *atypical* examples. Secondly, we make crucial design decisions to implement localization in the transformer architecture for language modeling task (as opposed to past work in image classification). This includes implementing `SeqTD` in synergy with key-value memory stores in the MLP layers of transformers as found by Nanda et al. (2023); Geva et al. (2021). Lastly, we explain the mechanism of isolation of memorization by dropout-based regularizers in Appendix F, which was an open question in prior work.

## 5.1 EMPIRICAL RESULTS

**Sequence-Tied Dropout Enables Learning Across Sequences.** In Figure 4(a), we compare the validation loss of sequence-tied dropout with standard training with and without repeated documents. Firstly, note that standard training with repeated sequences outperforms filtering them out. This indicates that the model does learn general capabilities from observing documents repeated multiple times in our setting. Next, we compare the standard trained models with *Sequence-Tied Dropout*. We observe that when evaluating without the memorization neurons, sequence-tied dropout achieves comparable validation loss to standard training with repetition.

**Dropping Out Memorization Neurons Forgets Memorized Examples.** In Figure 4(b), we show the loss on the repeated TinyStories documents. A standard trained model memorizes these sequences during training, achieving close to 0 loss on them. Dropping out the memorization neurons in `SeqTD` significantly increases the loss on these sequences, increasing the loss to roughly 66% of a standard trained model that does not memorize. Interestingly, the loss of sequence-tied dropout on memorized sequences increases later in training. We further examine this finding in Appendix F.

**Sequence-Tied Dropout Enables Superior Sequence Forgetting-Model Degradation Tradeoff.** In Figure 4(c), we compare `SeqTD`'s tradeoff between sequence forgetting and model degradation, compared to the post-hoc methods from Section 3. We show that `SeqTD` achieves the best tradeoff relative to post-hoc methods, achieving a higher loss on memorized sequences with significantly lower impact on validation performance. In particular, `SeqTD` achieves the closest sequence forgetting to a model trained without repeated sequences (Standard: No Rep), while significantly outperforming that model in validation loss.

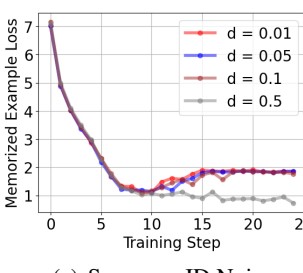 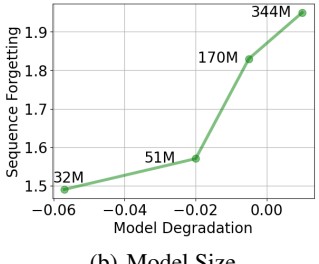 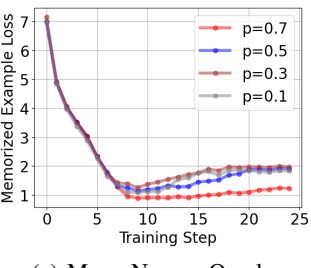

| (a) Sequence ID Noise | (b) Model Size | (c) Mem. Neuron Overlap |

Figure 5: **Practicality of** `SeqTD` (a) We study the impact sequence ID noise $d$, where a fraction of repeated documents have an inconsistent ID. `SeqTD` withstands small amounts of noise (up to 10%) (b) We examine the performance of `SeqTD` across model sizes, where we measure the model degradation as the change in validation loss relative to a standard model and the sequence forgetting as the loss on repeated sequences (c) We study the impact of the fraction of memorization neurons activated ($p$) on any given sequence.

## 5.2   PRACTICALITY OF `SeqTD`

There are two crucial requirements in deploying `SeqTD`: (a) accurate metadata that identifies repeated sequences and (b) the presence of memorization neurons which activate only on a subset of sequences. In this section, we study the sensitivity of `SeqTD` to these requirements.

**Noisy Metadata.**   `SeqTD` relies on consistent sequence IDs to activate the same memorization neurons across repetitions, requiring accurate metadata. However, large-scale pretraining corpora often contain noisy or incomplete metadata. To test robustness, we introduce random ID perturbations with probability  d. Our results show `SeqTD` remains effective with up to 10% noise but struggles at 50% noise, confirming that ID consistency is crucial for isolating memorization.

**Impact of Model Size.**   In Figure 5(b), we test the performance of `SeqTD` on a range of model sizes and find that it is capable of isolating memorization across model scales—as indicated by the comparably high losses on repeated sequences (relative to a normally trained model which attains nearly 0 loss). We find that model degradation (the increase in validation loss compared to a standard trained model of the same size) does grow as the model architecture becomes smaller. However, even on smaller models `SeqTD` outperforms post-hoc methods as shown in Figure 4(c). Thus, while model size plays a role in the success of `SeqTD`, the method has benefits in small models as well.

## 6   DISCUSSION

**Contribution.**   Our work addresses a significant open problem: Can memorization be disentangled from general model capabilities? In a controlled setting, we demonstrate that standard training can fail to do this—particularly in the practically impactful setting of typical sequences. However, we present a way to naturally promote disentanglement in pre-training by carefully leveraging the learning dynamics of memorization (`SeqTD`). In a small-scale setting, we verify that our method induces the isolation of memorization without compromising the learning of general capabilities. Moreover, we unveil the underlying mechanisms of `SeqTD`, which can inspire future techniques to promote isolation and modularity in LLM pretraining.

**Practical Considerations.**   There are practical considerations on the way to deploying `SeqTD` in real-world settings. Firstly, `SeqTD` can increase the size of model required for learning. Future work can examine memory efficient ways to implement the memorization neuron pool, for example finding ways to offload the bulk of their parameters to inactive memory by taking advantage of their sparsity. Secondly, `SeqTD` relies crucially on correct metadata to ensure that repeated sequences get routed to the right memorization neurons. Future work can examine efficient techniques for generating and correcting meta-data annotations based off of the semantic contents of sequences.

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

## A    PRIOR WORK ON UNDERSTANDING MEMORIZATION

There has also been significant interest in understanding the dynamics and mechanisms of sequence memorization. Tirumala et al. (2022); Carlini et al. (2019) showed that sequence memorization in LLMs often occurs before overfitting. Leybzon & Kervadec (2024); Chang et al. (2024a); Toneva et al. (2018) demonstrate that memorization often occurs in cycles of learning and forgetting throughout training. Geva et al. (2021); Dai et al. (2022) study the mechanistic implementation of memorization, finding MLP layers function as key-value memories. Huang et al. (2024) demonstrate that the decoding of memorized sequences may not be causally driven by a single memorization trigger, rather depending partially on certain likely next-token predictions. As a result, they argue that memorization can be highly "entangled" with general capabilities. In Section 3, we extend this finding, showing even when memorization significantly changes the models output (i.e. memorized sequences incur much lower loss than the validation set), identifying the neurons responsible for memorization can be infeasible.

## B    IMPLEMENTATION DETAILS OF TINYSTORIES TRAINING

**Implementation and Architecture.**   We use the nanoGPT library to perform standard pretraining of the models. We train a GPT-2-Medium like architecture with embedding dimension 1024 and a 4 times expansion in the MLP layer. We used 24 layers, the resulting model had approximately 344 M parameters.

Table 1: Hyperparameter Tuning for Standard Training

| Parameter | Values |
|---|---|
| Max Learning Rate | $\{$6e-5,6e-4,6e-3$\}$ |
| Weight Decay | $\{$1e-5,1e-3,1e-1$\}$ |
| Min Learning Rate | $\frac{\text{Max Learning Rate}}{10}$ |
| LR Decay Steps | Total Training Steps |

**Hyperparameter Tuning.**   We set the hyperparameters for our training as shown in Table 1. For parameters denoted in sets, we tuned over choices of these parameters relative to the validation loss. We also performed early stopping on the validation loss, but generally found that overfitting did not occur.

## C    IMPLEMENTATION DETAILS OF POST-HOC LOCALIZATION TECHNIQUES

We generally follow the methodology used in Chang et al. (2024b) and directly used their code as released online. We restrict our attention to their Hard-Concrete and Integrated Gradients methods presented in the papers.

**Hyperparameters: Hard Concrete.**   We tuned $\lambda$, the $\ell_1$ loss coefficient used in training the pruning mask $M$ over the values $\{100, 500, 1000\}$ on a tuning set of 5 sequences. Additionally, we tuned the number of pruning iterations in the range $\{1000, 2000, 4000\}$. The remainder of hyperparameters were set to the optimal values reported by Chang et al. (2024a). We tuned relative to the lowest validation loss achieved after dropping out the identified neurons.

**Hyperparameters: Integrated Gradients.**   For Integrated Gradients, the only hyperparameter was the number of IG steps. As a result, we set this to the value reported in the paper, 16.

**Dropout Procedure.**   Following the computation of mask scores by either Hard Concrete or attribution scores by Integrated Gradients, we sorted the neurons in each layer by these scores. Given a dropout parameter $r$, we dropped out an $r$ proportion of the neurons in each layer, as was performed in Chang et al. (2024a).

Table 2: Hyperparameter Tuning for Sequence-Tied Dropout

| Parameter | Values |
|---|---|
| Max Learning Rate | {6e-5,6e-4,6e-3} |
| Weight Decay | {1e-5,1e-3,1e-1} |
| Min Learning Rate | $\frac{\text{Max Learning Rate}}{10}$ |
| LR Decay Steps | Total Training Steps |
| $g$ | {0.7,0.9,0.95} |

## D  IMPLEMENTATION OF GRADIENT MASKING

We generally follow the implementation outlined in Cloud et al. (2024). We partition each MLP layer into memorization and generalization neurons. We tune this delineation of memorization and generalization neurons by the proportion of generalization neurons $g$. We additionally partition our dataset into examples seen once and the repeated examples. During training, we mask the gradients in each MLP layer such that the gradients from the repeated examples update only a the memorization block, whereas gradients of all other examples are routed to the generalization block.

**Hyperparameter Tuning.** We show the hyperparameters tuned for this method in Table 2. Hyperparameter denoted in sets are tuned relative to the validation loss *before* dropping out memorization neurons.

## E  IMPLEMENTATION OF SEQTD

Table 3: Hyperparameter Tuning for Sequence-Tied Dropout

| Parameter | Values |
|---|---|
| Max Learning Rate | {6e-5,6e-4,6e-3} |
| Weight Decay | {1e-5,1e-3,1e-1} |
| Min Learning Rate | $\frac{\text{Max Learning Rate}}{10}$ |
| LR Decay Steps | Total Training Steps |
| $g$ | {0.1,0.3,0.5,0.7} |
| $p$ | {0.1,0.3,0.5,0.7} |

**Implementation.** We partition the MLP neurons in each layer into shared neurons which are activated across all sequences, and memorization neurons of which only a fraction are activated on any given example (where the fraction is controlled by the memorization neuron dropout ratio $p$). We assign each sequence in pre-training data a sequence ID and use this as a seed to generate memorization neuron dropout masks. This enables us to ensure the consistency of dropout masks across repetitions of a sequence without precomputing and storing them in advance. We further emphasize that sequence IDs can be arbitrarily assigned (as long as repetitions of a sequence have the same ID). Thus, sequence ID can be generated "on the fly" for example by hashing the sequence.

**Experimental Details.** We train a GPT Medium model (same as all previous experiments), where 70% of MLP neurons are shared and the remaining 30% are allocated to the pool of memorization neurons. We emphasize that there are *far less* memorization neurons than total sequences. Thus, we do not assume each sequence can be allocated its own memorization neurons. We set the memorization neuron dropout ratio $p = 0.3$, but explore other choices in Section F. We train on the `TS-Repetition` dataset from Section 3.

**Model Architecture and Hyperparameterz.** We used the same model architecture as reported in Appendix B. We set the first $g$ fraction of neurons in each MLP as the "shared neurons" and left the remaining $1 - g$ fraction as the memorization neuron pool. We applied the dropout layer after the GeLU activation function, prior to the downprojection layer.

**Assignment of Sequence IDs.** We sequentially numbered the sequences in the TinyStories training set and use these indices as the sequence IDs.

**Hyperparameter Tuning.** In Table 3, we show the hyperparameter ranges tuned over for `SeqTD`. Hyperparameters denoted in sets were tuned over using the validation loss *when the memorization is dropped out.*

## F    HOW DOES SEQTD ISOLATE MEMORIZATION?

In this section, we investigate the mechanisms behind `SeqTD`'s ability to isolate memorization. Recall our hypothesis: having a set of neurons that (a) activate consistently across repetitions of a sequence and (b) activate on only a subset of other sequences would allow sequence-specific memorization to accumulate in these neurons and prevent it from being reinforced in shared neurons. Is this actually how `SeqTD` works?

**Testing the Role of Memorization Neurons.** We empirically test the hypothesis that memorization neurons in `SeqTD` are shielded from forgetting. For simplicity, we reran TinyStories pretraining setup with a single repeated sequence that is observed every 40 gradient steps. We track the training loss on this sequence for standard training and `SeqTD` (Figure 6). Recall that in `SeqTD`, the training loss on a sequence uses a forward pass with shared neurons and the sequence's assigned memorization neurons activated. Later in training, standard training continues to experience high-amplitude learning/forgetting cycles. `SeqTD`, on the other hand experiences less such fluctuations, maintaining a lower train loss on the repeated sequence. This provides evidence that memorization neurons have a shielding effect from the forgetting dynamics.

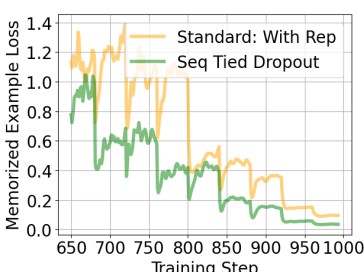

Figure 6: **Learning and Forgetting Dynamics of SeqTD.** We study a controlled setting where a specific TinyStories example is inserted every 10 gradient steps and compare the training loss on this sequence for standard training and `SeqTD`. We observe that `SeqTD` experiences lower loss and less forgetting spikes than standard training. This suggests that memorization neurons may provide insulated, long-term storage for repeated sequences.

**Why Can Memorization Neurons Tolerate Overlap?** We hypothesized that `SeqTD` insulates memorization neurons from interference and forgetting. However, this insulation is not perfect: as the number of neurons is *much smaller* than the number of sequences, there must be overlap between the memorization neurons assigned to different sequences. In Figure 5(c), we observe that when $p$ is set high (increasing the amount of overlap across sequences), the isolation effects of `SeqTD` do break down. For more moderate values of $p$, `SeqTD` is fairly robust. We note that it is not necessary to perfectly isolate memorization neurons from interference. Rather we must simply ensure that these neurons experience *relatively* less interference than shared neurons. In Theorem 3, we formalize this argument in a simplified setting, showing that different values of $p$ control the accumulation of memorization in shared versus memorization neurons.

**Unlearning in Shared Neurons.** In Sections 5.1 and 5.2, the loss on repeated sequences increases later in training. This suggests that some amount of memorization initially takes place in the shared neurons and is progressively "forgotten" later in training. We hypothesize that once the memorization neurons sufficiently"fit" the repeated sequences, additional observations no longer reinforce memorization in the shared neurons. Meanwhile, updates from other sequences remove memorization in the shared neurons, due to standard forgetting dynamics. This contrasts with standard training where any forgetting that occurs between observations of a sequence is reinforced throughout the *entire* model on subsequent encounters.

# G    ANALYSIS OF SEQ-TIED DROPOUT

## G.1    FORMALIZATION OF TRAINING PROCESS

**Architecture.**  For simplicity, we study the training dynamics of an MLP layer $f(x) = \mathbf{W}_{\text{proj}}\mathbf{W}_{\text{fc}}\mathbf{x}$, where $\mathbf{W}_{\text{proj}} \in \mathbb{R}^{d_{\text{h}} \times d_{\text{emb}}}$, $\mathbf{W}_{\text{fc}} \in \mathbb{R}^{d_{\text{emb}} \times d_{\text{h}}}$. Here, $d_{\text{emb}}$ refers to the embedding size of the model and $d_{\text{h}}$ refers to the number of hidden neurons in the MLP. Given a sequence $\mathbf{s}$, we consider that $f$ takes in the final position embedding of $\mathbf{s}$, which we denote $\phi(\mathbf{s})$ and directly outputs the logits of the next token (i.e. $\text{softmax}(f(\phi(\mathbf{s})))$ is a probability distribution over the next token in sequence $\mathbf{s}$.

For convenience, we will denote the hidden activations of sequence $\mathbf{s}$ as $\mathbf{z}(\mathbf{s})$. In our analysis, we will assume that the activation space of $\mathbf{z}(\mathbf{s})$ can be split into two subspaces $\mathbf{z}(\mathbf{s}) = [\mathbf{z}(\mathbf{s})_{\text{shared}} \quad \mathbf{z}(\mathbf{s})_{\text{mem}}]$. These components will correspond to our choice of shared and memorization neurons. We will additionally consider $\mathbf{W}_{\text{fc}}$ frozen throughout training and mainly study the training dynamics of $\mathbf{W}_{\text{proj}}$. Thus for convenience, we will also decompose $\mathbf{W}_{\text{fc}}$ into two column-blocks (corresponding to the shared and memorization neurons, respectively): $\mathbf{W}_{\text{proj}} = \begin{bmatrix} \mathbf{W}_{\text{proj}}^{\text{shared}} & \mathbf{W}_{\text{proj}}^{\text{mem}} \end{bmatrix}$

**Data Setup.**   We will treat our data as (embedding, next token) pairs. We consider we have a repeated sequence $\mathbf{s}^{\text{mem}}$ with corresponding next token $\mathbf{e}^{\text{mem}}$. Next, we will assume we have a large dataset of sequences seen only once during training $\mathcal{D}_{\text{once}} = \{(\mathbf{s}^{(1)}, \mathbf{e}^{(1)}), ..., (\mathbf{s}^{(N)}, \mathbf{e}^{(N)})\})\}$. For simplicity, we will consider the case where $\forall i \ \mathbf{e}^{(i)} \neq \mathbf{e}^{\text{mem}}$. Since we treat $\mathbf{W}_{\text{proj}}$ as frozen, we will also define $\epsilon_{\text{shared}} = \min \mathbf{z}(\mathbf{s}^{\text{mem}})_{\text{shared}}^{\top}\mathbf{z}(\mathbf{s}^{(i)})_{\text{shared}}$ and likewise that $\epsilon_{\text{mem}} = \min_i \mathbf{z}(\mathbf{s}^{\text{mem}})_{\text{mem}}^{\top}\mathbf{z}(\mathbf{s}^{(i)})_{\text{mem}}$. Intuitively, these quantities lower bound how similar the activations in the shared and memorization neurons are between the repeated example and any other example. For simplicity we will assume that the $||\mathbf{z}^{(\mathbf{i})}||_2 = 1$ for all $\mathbf{z}^{(\mathbf{i})}$ and that the parameter $||\mathbf{W}_{\text{proj}}||_2 < \frac{C_{\text{proj}}}{2}$ remains bounded throughout training. Finally we assume that the ouput embeddings $e$ are mutually orthogonal.

**Training Process.**   In standard training, we study the training trajectory (with learning rate $\gamma$) of minimizing the cross entropy loss with respect to the parameter $\mathbf{W}_{\text{proj}}$. We consider training with batch size 1.

## G.2    FORGETTING UNDER NORMAL TRAINING DYNAMICS

To begin, we introduce a result on the softmax with bounded inputs

**Theorem 1** (Softmax on $\ell_\infty$ bounded vectors). *Consider $x \in \mathbb{R}^d$ and suppose $x_\infty \leq C$. Then* $\max_i (\sigma(x))_i \leq \frac{e^{2k}}{d-1}$ *and* $\min_i (\sigma(x))_i \geq \frac{e^{-2k}}{d}$

*Proof.* $\sigma(x)_i = \frac{\exp(x_i)}{\sum_{j \in d} \exp(x_j)} \leq \frac{\exp(C)}{\exp(C)+(d-1)\exp(-C)} = \frac{\exp(2C)}{\exp(2C)+(d-1)} \leq \frac{\exp(2C)}{d-1}$. Likewise $\sigma(x)_i \geq \frac{\exp(-C)}{\exp(-C)+(d-1)\exp(C)} = \frac{\exp(-2C)}{\exp(-2C)+(d-1)} \geq \frac{\exp(-2C)}{d}$. $\qquad \square$

Given our assumption that $||\mathbf{z}^{(i)}||_2 = 1$ and the bounded parameter norm assumption $||\mathbf{W}_{\text{proj}}||_2 < \frac{C_{\text{proj}}}{2}$, it follows that $||\mathbf{W}_{\text{proj}}\mathbf{z}^{(i)}||_\infty \leq \frac{C_V}{2}$. By Theorem 1, we have that the entries of $\frac{\exp(-C_{\text{proj}})}{d_{\text{emb}}} \leq \sigma(f(\mathbf{z}^{(i)}) \leq \frac{\exp(C_{\text{proj}})}{d_{\text{emb}}-1}$, element wise. In the remainder of the theory, we denote $c_{min} = \frac{\exp(-C_{\text{proj}})}{d_{\text{emb}}}$ and $c_{max} = \frac{\exp(C_{\text{proj}})}{d_{\text{emb}}-1}$.

We will first show that the memorization of the repeated sequence $\mathbf{s}_{\text{mem}}$ is forgotten when we take intervening steps on non-repeated sequences $\mathbf{x}\mathbf{s}^{(i)}, ..., \mathbf{s}^{(i+n)}$. Formally, we have the following proposition. Formally, suppose that at after step $i$, we have just seen $\mathbf{s}^{(mem)}$. Then we will show that the logit $\mathbf{e}^{(\text{mem})}$ decreases during subsequent training steps $i$ through $i + n$. For this analysis, we will focus on the dynamics the shared neurons.

**Theorem 2** (Forgetting in Standard Training). *Suppose we take a gradient step on $\mathbf{s}^{(mem)}$ at gradient step $i$ and subsequently make gradient updates on non-repeated sequences $\mathbf{s}^{(i)}, ..., \mathbf{s}^{(i+m)}$. After the $m$ gradient steps, we have that $(\mathbf{e}^{mem})^{\top} f^{(i+m)}(\mathbf{z}^{mem}) \leq (\mathbf{e}^{mem})^{\top} f^{(i)}(\mathbf{z}^{mem}) - \gamma m \epsilon c_{min}$.*

*Proof.* Only the parameter $\mathbf{W}_{\text{proj}}$ changes throughout training, so we can restrict our attention to its dynamics. We have that the gradient of $\mathbf{W}_{\text{proj}}$ on the sequence-next token pair $(\mathbf{z}, \mathbf{e})$

$$\frac{\partial L}{\partial \mathbf{W}_{\text{proj}}} = (\mathbf{e} - \sigma(f(\mathbf{z}))\mathbf{z}^{\top}$$

Now let $\mathbf{W}_{\text{proj}}^{(i)}$ denote the parameter value of $\mathbf{W}_{\text{proj}}$ after the $i$-th observation. We have that

$$\mathbf{W}_{\text{proj}}^{(i+m)} = \mathbf{W}_{\text{proj}}^{(i)} + \gamma \sum_{j=1}^{m} (\mathbf{e}^{(j)} - \sigma(f^{(j+i)}(\mathbf{z^{(i)}}))\mathbf{z^{(i)}}^{\top} \tag{1}$$

where we will denote $f^{(j+i)}$ as the model with parameter $\mathbf{W}_{\text{proj}}$. Then, we have that the logit on the correct next token for memorized example $\mathbf{z}^{\text{mem}}$ is

$$(\mathbf{e}^{\text{mem}})^{\top} f^{(i+m)}(\mathbf{z}^{\text{mem}}) = (\mathbf{e}^{\text{mem}})^{\top} f^{(i)}(\mathbf{z}^{\text{mem}}) + (\mathbf{e}^{\text{mem}})^{\top} \gamma \sum_{j=1}^{m} (\mathbf{e}^{(j)} - (\mathbf{z}^{\text{mem}}) \sigma(f^{(j+i)}(\mathbf{z^{(i)}}))\mathbf{z^{(i)}}^{\top}(\mathbf{z}^{\text{mem}})$$

Now, since we have that the token embeddings are orthogonal, we can rewrite this as

$$(\mathbf{e}^{\text{mem}})^{\top} f^{(i+m)}(\mathbf{z}^{\text{mem}}) = (\mathbf{e}^{\text{mem}})^{\top} f^{(i)}(\mathbf{z}^{\text{mem}}) - (\mathbf{e}^{\text{mem}})^{\top} \gamma \sum_{j=1}^{m} \sigma(f^{(j+i)}(\mathbf{z^{(i)}}))\mathbf{z^{(i)}}^{\top}(\mathbf{z}^{\text{mem}})$$

Note that by the assumption of bounded norm for $\mathbf{W}_{\text{proj}}$. we have that $(\mathbf{e}^{\text{mem}})^{\top} \sigma(f^{(j+i)}(\mathbf{z^{(i)}})) \geq c_{min}$ (defined earlier). Note also the assumption that $\mathbf{z^{(i)}}^{\top}(\mathbf{z}^{\text{mem}}) \geq \epsilon \; \forall i$. This implies that

$$(\mathbf{e}^{\text{mem}})^{\top} f^{(i+m)}(\mathbf{z}^{\text{mem}}) \leq (\mathbf{e}^{\text{mem}})^{\top} f^{(i)}(\mathbf{z}^{\text{mem}}) - \gamma \sum_{j=1}^{m} \epsilon c_{min} \tag{2}$$

This immediately yields our desired claim. $\qquad\square$

Next, we will show that the seqTD accumulates memorization in the memorization neurons, as formalized in the following theorem. This theorem also crystalizes some key quantities relating to gradient interference. First of all, we see that the forgetting depends on the number of *further gradient steps* taken after seeing $\mathbf{s}^{\text{mem}}$. Secondly, we observe that the impact of forgetting dynamics is influcned by how *similar* the activation of neurons are amongst different examples: controlled by $\epsilon$. The first observation immediately suggests that if some neurons were activated less often, then those neurons would be effectively "store" more memorization.

### G.3 ANALYSIS OF SEQTD

**Theorem 3** (SeqTD Accumulates Memorization in Memorization Neurons)**.** *Consider training SeqTD, where the memorization neurons are activated on a $p$ fraction of non-repeated examples. We will assume that the model is trained from 0 initialization. Denote the MLP $f_{\text{mem-dropped}}$ as the model with memorization neurons dropped out and $f_{\text{gen-dropped}}$ as the model with the generalization neurons dropped out. Suppose that the model is trained for $N$ total steps and the repeated sequence $\mathbf{s}^{\text{mem}}$ is observed $k$ times. Then we have at the end of training*

*1.* $(\mathbf{e}^{mem})^{\top} f_{\text{gen-only}}^{(n)}(\phi(\mathbf{s}^{mem})) \leq \gamma k(1 - c_{min}) - \gamma(N - k)\epsilon_{\text{shared}} c_{min}$

*2.* $(\mathbf{e}^{mem})^{\top} f_{\text{mem-only}}^{(n)}(\phi(\mathbf{s}^{mem})) \geq \gamma k(1 - c_{max}) - \gamma(N - k)\rho\epsilon_{\text{mem}} c_{max}$

*where $c_{min}$ and $c_{max}$ are constants depending on an upper bound of the parameter norm of $\mathbf{W}_{\text{proj}}$.*

*Proof.* Our argument resembles the proof of Theorem 2, and we will rely on the intuition therein. For reference, we will write the gradients for the components of $\mathbf{W}_{\text{proj}}$ below.

$$\frac{\partial L}{\partial \mathbf{W}_{\text{proj}}^{\text{shared}}} = (\mathbf{e} - \sigma(f(\mathbf{z}))\mathbf{z}_{\text{shared}}^{\top}$$

and likewise

$$\frac{\partial L}{\partial \mathbf{W}_{\text{proj}}^{\text{mem}}} = (\mathbf{e} - \sigma(f(\mathbf{z}))\mathbf{z}_{\text{mem}}^{\top}$$

We will first examine $(\mathbf{e}^{\text{mem}})^{\top} f_{gen-only}^{(n)}(\mathbf{z}^{\text{mem}})$. At any point in training, recall that we can upper and lower bound the value $c_{min} \leq (\mathbf{e}^{(i)})^{\top}\sigma(f(\mathbf{z}^{\text{mem}})) \leq c_{max}$. As such, observe that $(\mathbf{e}^{(i)})^{\top}\sigma(f(\mathbf{z}^{\text{mem}}))$ received $k$ updates upper bounded by $\gamma(1 - c_{min})$ (from the $k$ obervations of $\mathbf{z}^{\text{mem}}$ and $(N - k)$ updates upper bounded by $\gamma\epsilon_{\text{shared}}c_{min}$ (from the remaining $(N - k)$ observations of the $\mathbf{z}^{(i)}$. This yields the desired claim for (1).

Now, for claim (2) observe that the component $(\mathbf{e}^{\text{mem}})^{\top} f_{mem-only}^{(n)}(\mathbf{z}^{\text{mem}})$ receives $k$ updates lower bounded by $(1 - c_{max})$ (again, from the $k$ observations of $\mathbf{z}^{\text{mem}}$, but only $p(N - k)$ updates from other observations, which can likewise be lower bounded by $\gamma\epsilon_{\text{mem}}c_{max}$ This immediately implies the desired claim in (2) $\qquad\square$

This theorem formalizes the notion that memorization "accumulates" in the memorization neurons when they are shielded from the interference of other sequences sufficiently. In our theory, the extent to which this occurs is dependent on two quantities (1) the fraction of *non-repeated* sequences for which the memorization neurons are active and (2) the similarity of activations of the repeated example and non-repeated example in the memorization neurons. Relative to algorithm design, however, we will generally only have control over $\rho$ and so we will consider $\epsilon_{\text{shared}} = \epsilon_{\text{mem}}$ out of convenience. Our analysis demonstrates that when $\rho$ is set appropriately low. Some calculation demonstrates that when $\rho < \frac{c_{min}}{c_{max}} - \frac{k}{(N-k)}(c_{max} - c_{min})$, then we will have a seperation in the logits of $\mathbf{s}^{\text{mem}}$ where the memorization neurons primarily contain the memorized example.