# OpenReview forum: "Disentangling Sequence Memorization and General Capability in Large Language Models"
_ICLR.cc/2025/Workshop/MCDC — MCDC @ ICLR 2025_

### Official Review · Reviewer_fh5y · 2025-02-27

**Rating:** 5
**Confidence:** 3
**Fit:** 1

**Summary:**

The authors first demonstrate that post-hoc localization methods (which try to identify and remove specific "memorization neurons") work poorly for typical text sequences, unlike for atypical content like random canaries. They show that simply enforcing isolation through gradient masking hinders cross-sequence learning and model performance. The authors propose Sequence-Tied Dropout (SeqTD), which partitions neurons into "shared" and "memorization" pools, with each repeated sequence consistently activating the same subset of memorization neurons. This approach leverages natural learning-forgetting dynamics: memorization accumulates in sequence-specific neurons while shared neurons focus on general linguistic patterns. In experiments using TinyStories, SeqTD successfully isolates memorization, enabling removal of repeated content without degrading overall model performance. The method is robust to moderate noise in sequence metadata (up to 10%) and works across various model sizes, though smaller models show more performance degradation. The paper provides theoretical analysis of how SeqTD works through learning-forgetting cycles and discusses practical implementation considerations like metadata accuracy and computational requirements.

**Reason For Giving A Higher Score:**

The paper solves a concrete LLM safety problem by working with the natural dynamics of memorization rather than fighting against them. The approach is theoretically grounded, empirically validated, and addresses a critical gap in the literature. The writing is clear and appropriately scopes both contributions and limitations.

**Reason For Giving A Lower Score:**

The topic is probably not the best fit for this workshop. The work is limited to small-scale experiments, leaving open questions about real-world viability. It requires additional metadata tracking during training that may be impractical for production systems. The computational overhead might prove prohibitive at scale, and the paper doesn't thoroughly explore this trade-off.

**Strengths And Weaknesses:**

Strengths:
Identifies a real problem in LLM training: disentangling memorization from capability+SeqTD handles typical text better than existing solutions
Clear empirical demonstration that isolation can occur naturally
Provides theoretical grounding for the approach
Method is robust to moderate (10%) metadata noise

Weknesses:
Limited to small-scale proof-of-concept (TinyStories)
Untested on multi-token memorization patterns
Requires sequence ID metadata during training
Smaller models show more degradation when using the technique
Resource overhead from maintaining memorization neuron pool

**Suggestions:**

Test on real-world copyrighted content rather than synthetic repeats
Quantify parameter/computation overhead vs standard training
Experiment with adaptive neuron pool sizes based on dataset size
Add ablation study on sequence consistency requirements
Compare with knowledge editing methods that modify trained models

---

### Official Review · Reviewer_Au7R · 2025-03-03

**Rating:** 7
**Confidence:** 3
**Fit:** 4

**Summary:**

The paper introduces a method to address the issue of verbatim memorization in LLMs, generally experienced during pretraining. Memorization has privacy and copyright concerns as the information can be extracted from the model, and hence, effective unlearning methods are important. The authors first show that memorized sequences are not naturally isolated to specific neurons during training, and current methods relying on this premise fail. The authors introduce a novel method, SeqTD, that explicitly disentangles memorization from generalization by partitioning neurons into “shared” and “memorization” groups. This approach ensures that memorization accumulates in a designated subset of neurons while preventing it from interfering with generalization. The authors evaluate SeqTD on a modified TinyStories pretraining setup, showing that it significantly outperforms post-hoc localization methods by allowing memorization to be erased without degrading the model’s generalization performance.

**Reason For Giving A Higher Score:**

The paper presents a novel idea to tackle a critical issue in language model training. It also puts forward a new perspective in tackling the issue, while presenting a wide range of experiments and analysis with promising empirical gains. The theoretical analysis further strengthens the claims and findings of the authors and creates a strong foundation for future research.

**Reason For Giving A Lower Score:**

The evaluation is primarily conducted on a controlled small-scale dataset, which raises concerns about the method’s scalability and real-world applicability. The method’s reliance on high-quality sequence metadata may further limit its practicality and scalability. However, as a start, I believe these issues are not very critical, but they still suggest that further work is needed before the approach can be broadly adopted.

**Strengths And Weaknesses:**

Strengths:
1. The training method and its basis are both novel. The perspective of having groups of neurons for generalized learning and memorization is interesting.
2. The paper presents a wide range of experiments with clear inferences.
3. Along with empirical studies, the authors also provide theoretical basis for their formulation of SeqTD and how it is able to isolate memorization.

Weaknesses:
1. The major weakness of the paper lies in the scale of the experiments - using a controlled setting with TinyStories. Real-world pre-training datasets would be significantly noisier and complex, and it is difficult to conclude what findings would extend to such scenarios.
2. SeqTD also relies on access to metadata, and also it being very accurate - this dependency can be problematic in more large-scale pre-training settings.
3. The added complexity of SeqTD creates an overhead that might not scale to the settings at which modern LLMs operate.

**Suggestions:**

1. Evaluate SeqTD on larger and more diverse datasets or with larger model architectures to better understand its scalability and generalizability.
2. Test the dependency of the method on metadata more rigorously and investigate possible substitutes for the same to cater to more realistic scenarios.
3. The paper could discuss computational overhead and potential memory constraints when applying SeqTD at large scales.

---

### Official Review · Reviewer_gtvR · 2025-03-03

**Rating:** 7
**Confidence:** 3
**Fit:** 3

**Summary:**

The paper "Disentangling Sequence Memorization and General Capability in LLMs" addresses the issue of memorization in large language models (LLMs), which poses risks for privacy and copyright. It finds that memorized sequences are not confined to specific neurons unless they are highly atypical, and existing methods struggle to isolate typical memorized sequences. The authors propose a new training method called Sequence-Tied Dropout (SeqTD), which isolates memorization to specific neurons using metadata to identify repeated sequences. SeqTD splits hidden-layer neurons into shared neurons and memorization neurons, allowing memorization to accumulate in specific neurons while preventing reinforcement in shared neurons. Empirical results show that SeqTD effectively isolates memorization and allows for unlearning repeated sequences without significantly affecting the model's performance on other data. The method can handle some noise in sequence metadata and works across different model sizes.

**Reason For Giving A Higher Score:**

The paper introduces a novel approach that appears to isolate memorization in (some smaller) language models. It provides thorough, albeit limited, empirical validation on a smaller dataset and restricted setting. Additionally, the paper discusses practical considerations and the potential for real-world deployment, and contains some analysis of robustness to metadata accuracy.

**Reason For Giving A Lower Score:**

The paper's limitations could have been discussed in more detail. Additionally, the rationale for the experiment design could have been better justified, such as the construction of canaries and the choice of 128 repetitions.

**Strengths And Weaknesses:**

Strengths:
- Novel Approach: To my knowledge, the Sequence-Tied Dropout (SeqTD) is a novel contribution. I find it a clever approach to use the metadata to partition hidden-layer neurons into shared neurons and memorization neurons.
- Empirical Validation: The paper provides thorough empirical validation, demonstrating that SeqTD effectively isolates memorization and allows for unlearning repeated sequences without significantly affecting the model's performance on other data.
- Practical Considerations: The paper discusses practical considerations such that accurate metadata i soften lacking and the impact of model size on SeqTD's effectiveness.
- Explores Robustness Assumptions: Experiments demonstrate that SeqTD can handle some noise in sequence metadata and works across different model sizes, indicating that is has potential for real-world deployment.

Weaknesses:
- Scalability: The paper would benefit from more extensive experiments on larger-scale models and diverse datasets to strengthen claims about the scalability and generalizability of SeqTD.
- Metadata Dependency: SeqTD relies on accurate sequence metadata to identify repeated sequences. The appendix has an interesting discussion about dynamic generation of metadata, but could benefit from exploring more robust metadata annotations noisy environments.
- Impact on Training Efficiency: The impact of SeqTD on training efficiency and computational resources is not thoroughly addressed. A detailed analysis of the computational overhead would provide a clearer picture of its practicality in large-scale settings.
- Canary Construction Process: The paper should provide a clearer rationale for the construction of canaries and how they represent real-world scenarios.
- Discuss the Realism of 128 Repetitions: Justify the choice of 128 repetitions and discuss whether this number is representative of real-world data. Consider the implications of using fewer or more repetitions and examine the limitations of this choice.
- Discussion of Limitations: The limitations of the experimental design could be discussed further.

**Suggestions:**

Suggestions for improvement
- Provide more discussion on how to robustly generate and maintain accurate sequence metadata.
- Discussion potential additional scaling experiments with larger datasets and larger language models.

There are some minor language errors that could easily be corrected, for example,
- Figure 1 Caption: "Conceptual {Intution} of SeqTD"
- Line 125: "these neurons {must also don’t contribute) to the model’s general capabilities"
- Line 291: "There are two {crucical} requirements in deploying SeqTD"

---

### Official Review · Reviewer_UMY4 · 2025-03-03

**Rating:** 7
**Confidence:** 4
**Fit:** 3

**Summary:**

This paper introduces SeqTD, a training paradigm to isolate sequence memorization. During training, it partitions the learning signal from each example to generalization and memorization components, where the memorization is isolated to specific memorization neurons by design. Results suggest that this effectively disentangles memorization from generalization on controlled settings.

**Reason For Giving A Higher Score:**

The method is well-motivated and intuitive.

**Reason For Giving A Lower Score:**

The experimental evaluation can be of larger scale and more complete in future work.

**Strengths And Weaknesses:**

Strenghts:
1. The paper is well-written, flows naturally.
2. The authors provided investigations on reasons why previous approaches, the post-hoc localization methods in particular, fail in some settings and well-motivated the method.
3. Experimental results in controlled settings demonstratred promising performance.

Weaknesses:
1. The method is currently evaluated in a small-scale controlled setting, the results can benefit from larger-scale and more practical settings in future work.

**Suggestions:**

1. Section 5, the main section to describe the method, can be extended to illustrate the method in more details rather than delaying details into appendix. Some details, e.g., how the metadata is used, how the neurons are splitted, can be explained and discussed more.
2. The proposed method can be evaluated across different settings, possibly both synthetic and real-world settings, for a more thorough evaluation of the method.

---

### Decision · Program_Chairs · 2025-03-06

**Decision:**

Accept

**Comment:**

The paper introduces a method to address the issue of verbatim memorization in LLMs, generally experienced during pretraining. During training, it partitions the learning signal from each example to generalization and memorization components, where the memorization is isolated to specific memorization neurons by design. Most of the reviewers agree that this is a good paper. However, we had some concerns about the relevance of the paper to the workshop's theme. In the end, we think that information localization can be thought of as  some form of modularity and hence recommend accepting the paper to the workshop given the strong reviews.